# Therapy of Chronic Viral Hepatitis: The Light at the End of the Tunnel?

**DOI:** 10.3390/biomedicines10030534

**Published:** 2022-02-24

**Authors:** Giorgio Maria Saracco, Alfredo Marzano, Mario Rizzetto

**Affiliations:** Gastro-Hepatoloy Unit, Department of Medical Sciences, University of Turin, 10126 Turin, Italy; alfredomarzano@yahoo.it (A.M.); mariohdv89@gmail.com (M.R.)

**Keywords:** chronic hepatitis B, chronic hepatitis C, chronic hepatitis D, interferon, entecavir, tenofovir, direct acting antivirals, bulevertide, lonafarnib

## Abstract

Chronic viral hepatitis determines significant morbidity and mortality globally and is caused by three main etiological actors (Hepatitis B Virus, Hepatitis C Virus, and Hepatitis D Virus) with different replicative cycles and biological behaviors. Thus, therapies change according to the different characteristics of the viruses. In chronic hepatitis B, long term suppressive treatments with nucleoside/nucleotide analogues have had a dramatic impact on the evolution of liver disease and liver-related complications. However, a conclusive clearance of the virus is difficult to obtain; new strategies that are able to eradicate the infection are currently objects of research. The therapy for Hepatitis D Virus infection is challenging due to the unique virology of the virus, which uses the synthetic machinery of the infected hepatocyte for its own replication and cannot be targeted by conventional antivirals that are active against virus-coded proteins. Recently introduced antivirals, such as bulevertide and lonafarnib, display definite but only partial efficacy in reducing serum HDV-RNA. However, in combination with pegylated interferon, they provide a synergistic therapeutic effect and appear to represent the current best therapy for HDV-positive patients. With the advent of Direct Acting Antiviral Agents (DAAs), a dramatic breakthrough has occurred in the therapeutic scenario of chronic hepatitis C. Cure of HCV infection is achieved in more than 95% of treated patients, irrespective of their baseline liver fibrosis status. Potentially, the goal of global HCV elimination by 2030 as endorsed by the World Health Organization can be obtained if more global subsidised supplies of DAAs are provided.

## 1. Therapy of Chronic Hepatitis B

The knowledge of Hepatitis B Virus (HBV) infection and its natural history is important to facilitate an accurate management of the chronic HBV infection (CHB). The major concepts are: (a) once infection occurs, HBV persists in almost all infected individuals, even after Hepatitis B surface Antigen (HBsAg) clearance; (b) the interplay between HBV and the host’s immune system is the driving force of the outcomes of HBV infection; and (c) a minor proportion of HBV carriers develop progressive liver disease and eventually die because of complications of cirrhosis and/or hepatocellular carcinoma (HCC). 

### 1.1. Epidemiology

About one third of the world’s population (2 billion people) have been infected by HBV, confirmed by anti-HBV antibodies *(past HBV infection)*, and approximately 240 million people are HBsAg positive *(overt carriers*), with a high variability of the endemicity levels. 

The prevalence of HBsAg is decreasing worldwide, mainly due to universal vaccination programs. Nevertheless, migration of HBV infected individuals from high to low endemic areas could further modify the overall picture of HBV infection, particularly with an increase of primary infection in unvaccinated adults. 

### 1.2. Virology and Pathogenesis

HBV is a small, enveloped, hepatotropic, non-cytopathic virus, which may persist in the infected cell without major alteration of cellular homeostasis. The small viral genome (3.2 kb) is a partially double-stranded, relaxed-circular (rc) DNA with a compact organization and four partially overlapped open reading frames encoding seven proteins. At variance with Hepatitis C Virus (HCV), the major steps of the viral life cycle, with the exception of reverse transcription of the pregenomic RNA into HBV-DNA, are mediated by cellular receptors, proteins, or enzymes. Thus, the interaction of the virus with the hepatocyte is pervasive and complex and could make it difficult to identify molecules acting exclusively on the virus machinery.

The interplay between the virus and the host’s immune system determines the outcome of HBV infection: (a) in *primary resolving infection*, the timely and synergistic response of both the innate and adaptive immune system achieves an effective control of the infection, inducing a robust adaptive T cell reaction (with cytolytic and noncytolytic antiviral effects) and the production by B cells of neutralizing antibodies, preventing the spread of the virus; (b) *in chronic infection*, HBV-specific T cell function appears to be impaired.

Albeit the major alterations of the immune response, the interaction between HBV and host immune system appears highly dynamic, and about 60% of chronic carriers will spontaneously achieve the control of HBV infection and resolve chronic hepatitis with transition to the phase of Hepatitis B e Antigen (HBeAg) negative infection [1].

### 1.3. Virological Categories

The natural history of chronic HBV infection has been schematically divided into five phases according to the two main characteristics, infection and hepatitis [2]:

HBeAg-positive CHB (previously termed “immune tolerant” phase) is characterised by serum HBeAg and very high levels of HBV DNA, while liver necroinflammation or fibrosis are minimal or absent, in absence of other causes of liver damage. 

HBeAg-positive CHB is characterised by the presence of serum HBeAg, high levels of HBV DNA, and moderate or severe liver necroinflammation, eventually associated with fibrosis. ALT is usually elevated. 

HBeAg-negative chronic HBV infection (previously termed “inactive carrier” phase) is classically characterised by serum anti-HBe, low (<2000 IU/mL) HBV DNA, quantitative HBsAg levels (<1000 IU/mL), and absent or minimal hepatic necroinflammatory activity, in absence of other causes of liver damage. In the classical phase 3, HBsAg loss and/or seroconversion may occur spontaneously in about 1–3% of cases per year. However, some subjects can be attributed to a “grey zone” (higher HBV DNA and/or quantitative HBsAg levels with persistent normal ALT). These anti-HBe-positive carriers without biochemical and histologic evidence of liver disease seldom progress to HBeAg-negative CHB and more frequently remain in this phase or show a further reduction of viral load [3].

HBeAg-negative CHB is characterised by the presence serum anti-HBe with persistent or fluctuating viremia. The liver histology shows necroinflamation and fibrosis. ALT fluctuates or is persistently elevated. Spontaneous disease resolution is rare.

The HBsAg-negative phase is characterised by serum negative HBsAg and positive anti-HBc, with or without detectable anti-HBs. Serum HBV-DNA is usually undetectable, whereas covalently closed circular DNA (cccDNA) can be detected in the liver. Viral induced liver disease is absent. Immunosuppression may lead to HBV reactivation in Occult B Infected patients (OBI) [4].

The treatment of HBV infection is indicated for phase 1 in order to induce anti-HBe seroconversion, and for phases 2 and 4, characterized by chronic hepatitis. However, phases 3 and 5, in absence of chronic hepatitis and significant staging, should be treated with antivirals only in cases of high risk (>10%) of clinical reactivation in immunosuppresed patients [5].

### 1.4. Current Scenario

Antiviral therapy is aimed to prevent progression of chronic hepatitis and cirrhosis or reactivation in immunocompromised patients. In immunocompetent patients, two different therapeutic approaches can be used to switch off disease activity: (1) *curative*, aimed to induce a change in the host–virus equilibrium, from pathogenic to nonpathogenic with a time-limited treatment able to obtain a persistent off-therapy control of HBV replication; (2) *suppressive*, based on the suppression of viral replication with continuous antiviral treatment.

Pegylated interferon (Peg-IFN) is the major player of the *curative strategy*; however, only 20–30% of the patients achieve a sustained virologic response (SVR), and it is contraindicated in the majority of immunocompromised patients.

In the suppressive strategy, the long-term (frequently life-long) treatment is based on nucleos(t)ide analogue (NA) drugs that are direct inhibitors of viral polymerase. Nevertheless, NA do not directly act on cccDNA and therefore do not promote the clearance of HBV infection; their discontinuation is associated with viral replication recurrence in the majority of patients. For this reason, at present, NA are usually maintained overtime in HBeAg-positive patients without anti-HBe seroconversion and in anti-HBe positive phase 3 patients without HBsAg clearance. 

In the last 20 years, many drugs have been used for the antiviral treatment of HBV: firstly Lamivudine (LAM), then adefovir dipivoxil (ADV) and telbivudine (TBV), and more recently, tenofovir disoproxil fumarate (TDF) and entecavir (ETV), the third generation of antivirals characterized by high antiviral efficacy and high genetic barrer, with consequent clinical improvement and reduction of liver transplantation need [6,7,8]. 

The impact of the antiviral therapy on HCC in cirrhotics has been long debated. Recently, better results have been described with ETV and TDF. A higher efficacy of TDF in the prevention of HCC has been recently reported, particularly by Asian authors, but it remains controversial [9,10].

### 1.5. Ongoing and Future Perspectives

There are multiple novel antivirals targeting different steps in the HBV life cycle currently in development. The aim of these drugs is the complete cure of the infection in analogy with HCV infection, and not only suppression. The final goal should be the control (functional cure) or the eradication of the infection (complete cure) [11] (Table 1). 

The different mechanisms of action of new anti-HBV therapies, mainly in phase II trials, are shown below [12]:

Identification of the Sodium Taurocholate Co-transporting Polypeptide (NTCP) expressed on the hepatocyte membrane has allowed the development of entry inhibitors, which are able to stop HBV and HDV infection of naive hepatocytes during the primary inoculation or the reinfection. Myrcludex B (Bulevertide) and Cyclosporine have exhibited this antiviral activity. Bulevirtide was recently registered in the US and Europe and is now available in clinical practice and in clinical trials as monotherapy or in combination with Peg-IFN.

Nucleocapsid assembly modulators are able to stop HBV core proteins, which are essential for HBV genome packaging. 

Post-transcriptional control inhibitors (RNAi or oligonucleotides) can directly target HBV transcripts and induce their degradation, resulting in gene silencing. 

HBsAg release inhibitors (Nucleic Acid Polymers) block the release of subviral HBsAg particles. Pilot studies performed in HBV and HDV patients using these drugs combined with TDF and Peg-IFN have been published in recent years with promising results [13,14,15]. However, all these studies described a hepatitis flare, clinically significant in some cases, preliminary to the therapeutic response. 

Therapeutic approaches aimed at suppressing cccDNA synthesis by small molecules are currently under way, but so far no clinical trials using these innovative drugs called cccDNA targets have been completed. 

Patients affected by chronic HBV hepatitis usually show immunologic dysfunction, suggesting a combined cure strategy. Immunomodulation can be induced by interferons, Toll-like receptor agonists, checkpoint inhibitors, and therapeutic vaccines. 

In conclusion, chronic HBV infection causes significant morbidity and mortality globally. Long term suppressive therapy with NAs showed a dramatic impact on the evolution of liver disease and liver-related complications. However, the need in many cases for prolonged therapy has stimulated the research of new strategies searching for a functional or complete cure of HBV disease, able to eradicate the infection in analogy with HCV. Unfortunately, HCV is an “easier” virus, without a nuclear replicative phase; the inhibition of cccDNA formation is the crucial challenge that should be addressed by the novel drugs, with an optimal safety profile similar to that of currently used NAs. 

## 2. Therapy of Chronic Hepatitis D

Peg-IFN is the only therapy for chronic hepatitis D (CHD) recommended by professional societies (not approved by Drug Regulatory Agencies); it has limited efficacy, and valid treatment of CHD has so far remained an unmet medical need [15].

The therapy for HDV infection is challenging due to the unique virology of the HDV [16]. The virus has a circular RNA genome of about 1700 nucleotides, which is too small to code for the complex viral polymerase and protease proteins that drive the autonomous replication process of ordinary viruses. It relies on the synthetic machinery of the infected hepatocyte for replication, which duplicates the viral genome through DNA-dependent RNA polymerases subverted to copy the viral RNA [17]; the corollary is that HDV cannot be targeted by conventional antivirals that are active against virus-coded proteins. Though the HBV infection required by HDV to become infectious could theoretically offer a target, treatment of HBV with ETV or TDF is of no avail, as the HDV requires from the partner virus only the HBsAg necessary to coat its virion and is not in need of its replicative machinery [18].

A second problem is the high potential infectivity of HDV on the background of a pre-existing HBV infection; end-titration experiments in HBsAg-susceptible chimpanzees have shown that an HDV-containing serum could transmit infection up to a 10^−11^ dilution [19]. Therefore, the persistence of HBsAg in patients who obtained an SVR may enable the late rescue of HDV still present in the liver at low levels but undetectable in serum with currently available HDV-RNA assays.

This raises the issue of how to determine the end point of therapy in CHD. Though the only robust end point is the clearance of the HBsAg, this is seldom achieved with current therapies. Therefore, in all CDH studies, the cardinal criterion of efficacy has been the clearance of HDV-RNA from serum, the so-called sustained viral response (SVR); in CHD, however, SVR is not an absolute end point of therapy, but rather the best that can be presumed in clinical practice [20]. Based on a small study showing an association of HDV decline with survival benefit [20], a ≥2-log reduction in serum HDV-RNA from baseline was proposed as initial treatment efficacy in clinical trials for CHD [21]. Subsequent studies used this log reduction as a therapeutic end point, making it difficult to interpret the results and especially the comparison with studies that adopted viral clearance as their primary treatment end point [22].

Current therapeutic efforts are directed to deprive the HDV of HBsAg functions critical to its life cycle [18]. Three therapeutic strategies are currently being evaluated. As the HBsAg enters hepatocytes through the NTCP expressed on the cell membrane [23], drugs that interfere with the NTCP may prevent access of the HDV into the cells. As the assembly of HDV virions requires the farnesylation by the host of the large HD antigen of the virus [24], interference with this cellular process may lead to the disruption of viral assembly [25]. As the HDV needs to encapsidate in the HBsAg coat for discharge into the blood, nucleic acid polymers (NAPs) that appear to prevent the synthesis of subviral HBsAg particles may prevent the export of the HD virion to the blood [26].

### 2.1. Nucleic Acid Polymers

The NAP REP 2139-Ca given to 12 CHD patients for 15 weeks as monotherapy, followed by add-on Peg-IFN for 15 weeks and then Peg-IFN monotherapy for another 33 weeks, led at the end of therapy to undetectable HDV-RNA in 7 patients and the loss of HBsAg in 4 patients [27]. These results were maintained after a 3.5 year follow-up [14]. These preliminary data of REP 2139/Peg IFN in a small series are promising, but further studies are needed to confirm the impressive response rates.

### 2.2. The Farnesyl-Transferase Inhibitor Lonafarnib

In a pilot study, the farnesylation inhibitor Lonafarnib (LNF), given orally, decreased serum HDV-RNA levels, but was aggravated by gastrointestinal side effects [25]. Subsequent studies have used LNF in combination with the cytochrome P450 3A4 inhibitor Ritonavir to permit a lower dose of LNF while preserving its antiviral activity. In the LOWR-2 study [28], HDV-RNA became undetectable in 5 of 13 patients given LNF 50 mg bid with Ritonavir 100 mg bid for 24 weeks. In the LIFT-HDV study, serum HDV-RNA became undetectable at the end of treatment in 11 of 26 patients, given LNF and Ritonavir together with Peg-IFN lambda at weekly doses of 180 µg for 24 weeks [29]; IFN lambda is credited to have fewer side effects than IFN alfa. In the ongoing phase 3 D-LIVR study, LNF plus Ritonavir is combined with Peg-IFN lambda for 48 weeks. In light of the need for long-term therapies, the side effects of LNF, though mitigated by Ritonavir, might remain a concern, particularly when added to those of Peg-IFN.

### 2.3. Bulevertide

Bulevertide (BLV), formerly Myrcludex B, a myristolated synthetic lipopeptide corresponding to the preS1 sequence of the HBsAg [30], is used to block the engagement of the HBsAg of the HDV with the NTCP in order to prevent the de-novo infection of yet uninfected liver cells, with the aim to eliminate all HDV-infected hepatocytes and recolonize the liver with regenerating HDV-free cells. It is administered daily by the subcutaneous route and is generally well-tolerated despite a dose-dependent bile acid increase. On 31 July 2020, the European Medicines Agency has afforded a conditional marketing authorization to BLV under the trade name Hepcludex, with a recommended dose of 2 mg daily [31].

Preliminary data were reported in abstract form in the study MYR 202 and MYR 203. In MYR 202 trial [32], HDV RNA decreased by ≥2 Log or became undetectable by the end of therapy in 46–77% of the 90 patients given TDF for 12 weeks followed by BLV 2, 5 or 10 mg plus TDF for 24 weeks, and then by TDF alone for 24 weeks; the best response was seen in the group given BLV at a 10 mg dose. However, only 7–10% of patients maintained the HDV RNA response in the follow-up.

In the MYR 203 study [33,34], 90 patients were entered in six groups of 15 patients each and treated for 48 weeks. After 24 weeks of post-therapy follow-up, HDV-RNA was undetectable in 8 (53%), 4 (27%), and 1 (7%) of the patients given the combination of Peg-IFN and 2, 5, or 10 mg BLV, respectively; HDV-RNA was undetectable in 1 (7%) of the patients given 2 mg BLV monotherapy, in 3 (33%) of those given 10 mg BLV and TDF, and in none of the patients given Peg-IFN alone. ALT remained normal in 7and 5 of the 15 patients treated with the two combinations, and in 3 patients given 2 mg of BLV. The HBsAg became undetectable in 4 of the 15 patients treated with the combination using 2 mg of BLV.

These encouraging results have led to the design and implementation of two long-term studies that are ongoing, one of finite therapy with Peg-IFN and BLV (MYR-204 Phase 2b Study) and one of chronic therapy with BLV alone (MYR-301 Phase 3); data at the 24 weeks interim have been reported for both studies. In the MYR-204, 25, 50, 50, and 50 patients are treated with Peg-IFN alone, BLV 2mg + Peg-IFN, BLV 10 mg + Peg-IFN, and BLV 10 mg, respectively; undetectable HDV-RNA is the primary end-point [35]. At week 24 of therapy, serum HDV-RNA was undetectable in 13, 24, 34, and 4 patients, respectively, and ALT had normalized in 13%, 30%, 24%, and 64%, respectively. In the MYR-301, 49 patients were treated with BLV 2 mg, 50 patients with BLV 10 mg, and 51 were left untreated; the primary endpoint is the combination of HDV-RNA undetectable or decreased by ≥2 log IU/mL from baseline with ALT normalization [36]. This was achieved in 6%, 53%, 38%, and 6% of the patients, respectively.

Interim data are also available from patients recruited in a compassionate study of BLV in France [37]. Seventy-seven patients treated with BLV 2 mg alone and sixty-eight treated with BLV 2 mg in combination with Peg-IFN have been considered in a per-protocol analysis at month 12 of therapy; 39% of the first and 85% of the second had HDV-RNA undetectable and serum ALT had normalized in 48.8% of the first and 36.4% of the second. These results are outstanding but require confirmation in a properly designed prospective randomized study in patients with homogeneous demographic and clinical features using a common standardized procedure to detect HDV-RNA.

In conclusion, BLV and LNF in combination with Peg-IFN provide a synergistic therapeutic effect and appear to represent the best therapy for CHD patients that can tolerate Peg-IFN.

In patients who cannot tolerate Peg-IFN, long-term BLV monotherapy may provide an alternative. Though less active against the HDV than the combinations, it has driven good biochemical responses and has been generally well tolerated; BLV monotherapy would seem the only viable option for the many HDV cirrhotics who are at risk with Peg-IFN. 

Prolonged treatments raise the concern of the safety of LNF, especially in association with the poorly tolerated Peg-IFN alfa. Peg-IFN lambda might provide an alternative, as it is credited with fewer side effects than Peg-IFN alfa.

## 3. Therapy of Chronic Hepatitis C

HCV affects about 71 million people worldwide [38], leading to liver cirrhosis and HCC in many cases; moreover, the infection is associated with several nonhepatic diseases with an overall mortality related to the extrahepatic complications of 580,000/year [39]. The advent of direct-acting antiviral (DAA) treatment, including RNA-dependent polymerase inhibitors (anti-NS5B), protease inhibitors (anti-NS3/4A), and anti-NS5A inhibitors, has significantly improved the therapeutic success for HCV infection, providing a simplified approach for global HCV elimination by 2030 as endorsed by the World Health Organization [40]. According to the European Association for the Study of the Liver (EASL), the aim of treatment is to cure HCV infection to prevent the complications of HCV-related liver and extrahepatic diseases, including liver necroinflammation, fibrosis, cirrhosis, decompensation of cirrhosis, HCC, and severe extrahepatic manifestations, to improve quality of life, and to prevent onward transmission of HCV [41]. Such beneficial effects have an impressive impact on the reduction in mortality, irrespective of the baseline liver fibrosis [42,43,44]. Cure of HCV infection is defined by the achievement of the sustained virological response (SVR), i.e., undetectable HCV-RNA in the serum of patients 12 or 24 weeks after the end of antiviral treatment; this surrogate end point has been validated by observing the very low rate of post-SVR relapse and is also a surrogate marker of improved liver-related morbidity and mortality [45]. Currently, there are two approved pangenotypic DAA regimen available, namely Sofosbuvir and Velpatasvir (SOF/VEL), as well as Glecaprevir and Pibrentasvir (G/P). While both regiments are effective in inducing SVR rates beyond 95% in most scenarios, only SOF/VEL is approved to treat decompensated HCV cirrhosis patients [46,47].

This review will evaluate the long-term benefits provided by DAA on hepatic and extra-hepatic outcomes.

### 3.1. Liver Outcome

#### 3.1.1. Compensated Cirrhosis

In the Interferon era, regression of fibrosis in HCV patients with cirrhosis was documented after SVR by pre- and post-therapy liver biopsies in 61% of patients [48]. It is reasonable to assume that with the advent of DAA, this histological benefit will be even more frequent. However, due to the lack of post-SVR liver biopsies, we have no current solid data regarding the long-term histologic outcome of cured cirrhotic patients. According to EASL, noninvasive scores and liver stiffness measurement (LSM) by transient elastography (TE) and other elastography methods are not accurate in detecting fibrosis regression after SVR and their routine use is currently not recommended [49]. It is well known that in cirrhotic patients with HCV, DAA-induced SVR decreases the risk of liver-related complications as well as all-cause mortality [50,51]. SVR is associated with a decrease in the incidence of liver-related events in the vast majority of cirrhotic patients [52,53]; in particular, DAA-induced viral clearance results in a significant reduction of incident HCC [50,54,55], while the issue regarding the recurrence rate of HCC in patients achieving SVR is still matter of debate [56,57,58].

Cirrhotic patients achieving SVR by DAAs show a progressive decrease in portal pressure during follow-up, reducing the incidence of decompensation events [52,59,60,61]. However, clinically significant portal hypertension (CSPH) may persist in a significant proportion of them [62,63,64], and several noninvasive tests (NITs) are currently used to stratify cured patients in order to better individuate patients at risk for liver decompensation [65]. According to the recent EASL guidelines [49], in successfully treated HCV-positive cirrhotic patients, LSM by TE could be helpful to refine the stratification of the residual risk of liver-related complications, even though cured cirrhotic patients should continue to be monitored for HCC by abdominal ultrasound examination ± alphafetoprotein assay every 6 months irrespective of the results of NITs. This stringent recommendation is based upon the finding that HCC is the most frequent liver-related complication after SVR [66]. The need for assessing predictive factors of HCC occurrence in order to individuate HCC surveillance has prompted many hepatologists to look for NITs, both before and after therapy, but currently no specific NITs or algorithms combining different risk factors have been officially validated [66].

#### 3.1.2. Decompensated Cirrhosis

Patients with decompensated cirrhosis may benefit from antiviral treatment with DAAs, even though most clinical trials [67,68,69,70,71,72] showed a significant decrease in SVR rates among decompensated cirrhotics. However, liver function improves as confirmed by amelioration both in Child-Turcotte-Pugh (CPT) classification and Model for End- Stage Liver Disease (MELD) scores in a significant proportion of patients [73,74,75,76,77]. Whether such benefit is durable over the long-term is still matter of debate [78,79,80,81]. Moreover, such improvement is rarely found among patients with severely impaired liver function at the start of therapy [82]; for this reason, it is paramount to establish a pretreatment scoring system based upon NITs able to individuate patients in whom therapy could be futile or harmful. International guidelines [83] suggest not to treat patients with an MELD score >20 because this particular subset of patients may be delisted from liver transplantation due to transient clinical improvement while still being at risk of lethal complications (the so-called “MELD purgatory”). However, this threshold seems to be inaccurate as recent studies [81,82] showed that many patients with lower MELD scores may not obtain significant clinical benefit over the long term despite SVR.

A predictive scoring system (the BE3A scoring system) adopting five pre-therapy features (BMI, lack of encephalopathy, lack of ascites, ALT > 60 IU/L, and albuminemia > 3.5 g/dL) was recently published [84]; patients with high scores had the highest chances of achieving CPT class A, but they represented less than 5% of the considered patients. Conversely, patients with baseline low scores had less than 25% of chances of achieving CPT class A suggesting that Orthotopic Liver Transplant (OLT) would have been the best solution rather than antiviral therapy. Further studies are needed in order to validate NITs or algorithms using combinations of NITs able to define the point of no return in this particular category of patients.

#### 3.1.3. Liver Transplant Setting

The advent of DAAs has revolutionized HCV treatment in the liver transplant (OLT) setting. Therapy of HCV infection pre-OLT in patients awaiting liver transplantation has two main aims: preventing liver graft infection after OLT and improving liver function before transplantation. According to the EASL guidelines [45], patients without HCC awaiting OLT with a MELD score < 18–20 should be treated prior to liver transplantation while patients with a MELD score > 18–20 should be transplanted first, and HCV infection should be addressed after OLT. Only patients with an expected waiting time on the transplant list >6 months should be treated before transplantation. Pre-OLT therapy seems to be an appropriate strategy especially in those areas where the average age of the donor exceeds 60 years [85], with a higher risk of graft dysfunction immediately after OLT [86]. Early allograft dysfunction (EAD) shows a negative clinical impact on graft and patient survival, often involving other organs such as kidneys [87]. For this reason, negativization of viremia by pre-OLT antiviral therapy should be a priority in order to prevent graft infection at reperfusion and to reduce EAD incidence [88].

However, unpredictable waiting time, antiviral therapy duration, risk of patient death on the list, and higher rates of SVR in transplant recipients compared with decompensated cirrhotic patients induce clinicians to treat infection after OLT. In fact, treatment following liver transplantation has greatly ameliorated post-OLT survival [89,90], with an overall SVR rate > 95%. Thanks to recent studies [91,92], international recommendations [45] regarding this hot issue were finally drawn, suggesting pre-OLT therapy for patients without HCC with a MELD score ≤ 20, while DAAs after OLT are cost-effective in patients with a MELD score > 20.

#### 3.1.4. Extrahepatic Manifestations Outcome

A causal relationship between HCV infection and extrahepatic manifestations (EM) (in particular, cryoglobulinemic syndrome, Non-Hodgkin’s Lymphoma (NHL), diabetes mellitus (DM), cardiovascular, neurological, and kidney diseases) was proven [93] and current guidelines [41,94] strongly recommend DAA therapy in HCV-positive patients with clinically significant extrahepatic manifestations. The advent of such treatment has significantly decreased the overall prevalence of HCV-related EM [93] even though the conclusive amount of the beneficial effects has not yet been completely assessed due to the short follow-up and the controversial results reported so far.

#### 3.1.5. Mixed Cryoglobulinemia

Mixed cryoglobulinemia (MC) is a B-cell lymphoproliferative disorder and consists of polyclonal IgG with monoclonal or polyclonal IgM and rheumatoid factor activity which precipitate when the temperature is below 37 °C, determining a small-vessel systemic vasculitis [95]. Symptoms of MC vasculitis are also known as MC syndrome, which is characterized by palpable purpura, weakness, and arthralgias and by several organs and tissue involvement such as skin, kidney, nervous, cardiovascular, and digestive systems [96]. MC is present in about 40–60% of HCV-positive patients and up to 30% of them show symptomatic cryoglobulinemic vasculitis (CV) [97,98,99]; 5–10% show an evolution to NHL [97,100,101] with a reduced life expectancy [95,102].

The introduction of DAAs has led to a complete or partial remission of the CV-related manifestations in the vast majority of patients with only a minority of nonresponders/relapsers [103,104,105,106,107,108,109,110,111,112,113]. However, long-term complete eradication of MC is observed in only 29–66% of patients [93], reflecting a B-cell clonal expansion persistence [114,115,116]. From a clinical point of view, this persistence is associated to the maintenance or recurrence of CV-related symptoms in a nonnegligible minority of patients, in particular among those with renal and/or neurological involvement [107,108,109,110,111,112,113].

When compared with the Interferon (IFN) era, DAA-induced clinical results regarding CV seem to be less definitive, suggesting that IFN could be better than DAAs on clinic-immunological outcomes due to its antiproliferative and immunologic activity. For this reason and due to the risk of NHL occurrence, long-term follow-up of patients with MC achieving SVR is mandatory.

#### 3.1.6. B-Cell Non Hodgkin’s Lymphoma

NHL comprises different lymphoproliferative disorders, but the link between HCV and haematological neoplasias was only proven for specific B-cell origin malignancies (B-NHL) [117].

B-NHLs most frequently associated with HCV are the marginal zone lymphoma, lymphoplasmacytic lymphoma, and diffuse large B-cell lymphoma [118]. The pathophysiology of such correlation is still matter of debate, but it is likely that continuous and sustained stimulation of lymphocyte receptors by viral antigen, viral replication in B cells, amd genetic alterations play a significant role in the lymphomagenesis [119]. The prevalence of HCV-associated B-NHL is variable, ranging from 20% in Italy to 6% in Europe (with the exclusion of Italy) [93]. Several reports have shown the regression of HCV-related indolent B-NHL with IFN-free antiviral therapy with very high rates of progression-free survival at 1 year [120,121,122,123,124,125,126]. These impressive results associated to the elevated tolerability and safety of DAAs have prompted many hepatologists to start antiviral treatment prior to administration of chemotherapy also in patients with aggressive B-NHL in order to neutralize B-NHL trigger and decrease the risk of relapse.

#### 3.1.7. Neurologic Manifestations

Neurologic and neuropsychiatric manifestations may occur in about 15–45% of HCV-positive patients [127]; peripheral neuropathy (PN) characterized by sensory loss and motor weakness [128] is mainly due to CV which induces a neural ischemic damage by occluding the epineural arterioles and small vessels. In contrast to PN, symptoms of central nervous system impairment (anxiety, depression, fatigue, attention and memory deficits, sleep disturbances, and confusion) are rarely due to CV; a direct neurotoxic effect has been hypothesized thanks to the finding of brain neuro-invasion by HCV and intrathecal replication [129,130]. Studies published so far [106,109,111,115,131,132,133,134,135] on the impact of DAA-based treatment on neurologic disorders have shown a significant reduction of neuropathic pain, even though PN demonstrated a lower clinical response [127] compared to cutaneous and articular manifestations (30–70% vs. 75–100%). A beneficial effect on neuropsychiatric and cognitive affections related to HCV infection was also found [136,137], with the improvement of fatigue, sleep disturbances, vitality, mental component summary, general health, and in the activity of cerebral cortex profiles.

#### 3.1.8. Chronic Kidney Disease

HCV-positive patients have a higher risk of chronic kidney disease (CKD) than uninfected patients [138,139,140], and various mechanisms have been reported to explain this difference; the most frequent is a membranoproliferative damage induced by CV [93,141] but CV-free membranoproliferative glomerulonephritis, membranous nephropathy, and tubulointerstitial injury are also described [142,143]. Once it is established that the kidney is a relevant target of the extrahepatic activity of HCV, it is reasonable to assume that achievement of SVR may reduce the incidence of “de novo” kidney diseases and improve concomitant nephropaties. All approved DAAs can be used in patients with mild-to-moderate renal impairment [144], and recently the exclusion of sofosbuvir-based therapies in patients with severe renal impairment has been removed [145]. The great majority of studies published so far [146,147,148,149,150,151,152,153] have shown that DAAs are effective in lowering the risk of kidney disease in HCV-positive patients and in stabilizing/improving renal function in patients with CKD, even though the long-term impact on kidney survival is still largely unknown [93].

#### 3.1.9. Cardiovascular Diseases

There is robust evidence that HCV infection is associated with cardiovascular diseases (CVD) [154,155,156], exerting its detrimental effect through direct (inducing a proinflammatory and profibrogenic environment) and indirect (determining metabolic co-morbidities such as insulin resistance (IR) and DM) mechanisms [157].

The advent of DAAs has led to a significant reduction of the risk of cardiovascular events [158,159,160,161], and this strong clinical impact is still maintained even when potential confounders such as liver fibrosis are considered [162,163]. This effect is probably due to the decrement in atherosclerosis as reported by Italian authors in [164,165,166].

The SVR induced by DAAs on CVD is still a matter of debate; larger prospective studies with longer follow-ups are needed before drawing definite conclusions regarding the long term benefits of viral clearance.

#### 3.1.10. Diabetes Mellitus

Several reviews and meta-analyses [167,168,169] have shown a higher incidence and prevalence of DM in patients with HCV than in controls. According to some authors [170,171], the virus can interfere with insulin signaling, eventually inducing alterations in glucose homeostasis. However, recent data suggested a direct role of HCV [172] by inducing death of pancreatic beta cells and upregulating several hepatokines known to cause insulin resistance. Achieving SVR by DAAs is associated with a reduced incidence of DM and a significant improvement in glycemic control among diabetic patients [93,173,174,175,176,177] (Table 2, [178,179,180,181,182,183,184,185,186,187]). Moreover, this beneficial effect seems to have a clinical impact on DM-related complications [161].

Howeverd the influence of DAA-induced SVR on the long-term outcome of DM in diabetics remains largely unknown; few studies with long-term follow-up addressing this issue have been published, and they present conflicting results [182,187]. To explain this discrepancy, it is important to note that the glycometabolic control may be affected by viral clearance and genetic and lifestyle-related factors, such as dietary habits, physical activity, and therapeutic adherence, which are prone to change over the long term.

In conclusion, with the introduction of DAAs, the great majority of treated patients definitively cleared the virus and achieved a permanent recovery, with a significant improvement in liver-related outcomes and extra-hepatic manifestations. However, unsolved issues remain, including the role of DAAs in patients with decompensated advanced liver disease, the management of patients not responding or relapsing after treatment with DAAs, and the persisting risk of HCC in cirrhotics after achieving SVR. Moreover, due to the limited worldwide access to healthcare, the majority of patients remain untreated and undiagnosed, and will develop liver complications in the future. For this reason, international organizations and high-income countries should help low-income countries to prioritize screening policies and access to DAA treatment.

## Figures and Tables

**Table 1 biomedicines-10-00534-t001:** Goals and definitions for HBV therapies.

Response	Blood				Liver
	ALT	HBV DNA	HBsAg	Anti-HBs	cccDNA
Virologic	normal	undetectable	detected	undetectable	present
Biochemical	normal	N/A	detected	undetectable	present
Functional cure	normal	undetectable	undetectable	detected	present
Complete cure	normal	undetectable	undetectable	detected	undetectable

ALT: alanine aminotransferase; anti-HBs: anti-HBsAg antibodies, cccDNA: covalently closed circular DNA, HBsAg: Hepatitis B surface Antigen.

**Table 2 biomedicines-10-00534-t002:** Studies recruiting ≥100 HCV-positive diabetic patients and reporting significant glycometabolic amelioration after successful therapy with DAAs.

Author, Year, (Ref.)	Type of Study	Design	N.	Mean FPG Change (*p*)	Mean HbA1c Level Change (*p*)	Follow-Up
Hum et al., 2017 [178]	Observational	Retrospective	2180	Not determined	−0.37% (0.03) *	48 weeks
Dawood et al., 2017 [179]	Clinical trial	Open label	378	−23.4 mg/dL (N.A)	−0.45% (N.A.)	12 weeks
Ciancio et al., 2018 [180]	Observational	Prospective	101	−18.0 mg/dL (0.002)	−0.5% (<0.001)	12 weeks
Gilad et al., 2019 [181]	Observational	Retrospective	122	Not determined	−0.6% (0.001)	1.5 years
Li et al., 2019 [182]	Observational	Retrospective/Prospective	192	Not determined	−2.3 (<0.001)	24 weeks
Boraie et al., 2019 [183]	Observational	Prospective	116	−8.4 mg/dL (0.01)	0.9% (0.008)	12 weeks
Andres et al., 2020 [184]	Observational	Retrospective	310	Not determined.	−0.27% (0.014)	1.6 years
Wong, 2020 [185]	Observational	Retrospective	937	Not determined	−0.39% (<0.0001)	12 months
Zied, 2020 [186]	Observational	Prospective	100	−107 mg/dL (0.005)	−0.41% (0.003)	12 w
Ciancio, 2021 [187]	Observational	Prospective	141	−15 mg/dL (0.001)	−0.7% (0.003)	44.5 months

DAAs = Direct Acting Antiviral Agents; HCV = Hepatitis C Virus; ref. = reference; N. = Number; HbA1c = Haemoglobin A1c; N.A. = not available; * = −0.13%, *p* = 0.01 when adjusted by multiple regression analysis.

## Data Availability

Not applicable.

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
