# Peer review of "Therapy of Chronic Viral Hepatitis: The Light at the End of the Tunnel?"

_biomedicines, 2022, doi:10.3390/biomedicines10030534_

Round 1

Reviewer 1 Report

The manuscript entitled THERAPY OF CHRONIC VIRAL HEPATITIS: THE LIGHT AT THE END OF THE
TUNNEL? is an interesting review article focusing on current trends in the treatment of chronic viral hepatitis B, C and D.
After a brief introduction to each type of chronic hepatitis, the prospects for treatment and the present are compared.
I think that such material will be interesting for clinicians as well as for basic scientists and their research.
It is written in good language and style, modern literature is used.

Author Response

Thank you for the interest and appreciation

Reviewer 2 Report

Saracco, et al. summarize current therapeutic status of chronic viral hepatitis due to HBV, HDV, and HCV, as well as extrahepatic manifestations by HCV. I found little novel description about HBV although they well-describe about HDV, HCV, and HCV-extrahepatic manifestation. They showed several anti-HBV drugs, but “the light at eth end of the tunnel” still can not to be seen. Therefore, authors can delete this part.

Minor

  1. HBV introduction line 13

I cannot agree with the reason why the prevalence of HBsAg is decreasing is “ because of the improvement of socioeconomic conditions”. Please re-consider this description.

  1. Line 17

“not directly cytopathic virus” can be substituted to “non-cytopathic virus”.

Author Response

"The light at the end of the tunnel" shows a question mark at the end of the title, meaning that it is a question not an assertion.  This is surely true regarding HBV and HDV therapy as a final and conclusive curative treatment for both chronic viral infections is still lacking.  However, our review is aimed to provide readers with the most updated data regarding new drugs and therapies, even though they have not yet been validated.  Anyway, according to your suggestion we have shortened the "Current scenario" section by erasing a paragraph (Therapy, Current scenario, page 6, from line 7 to line 16). 

Minor comments

1) Introduction, line 13.  Your suggestion has been met and the "improvement of socioecononomic conditions" as reason for the decrease of HBV prevalence has been erased.

2) Introduction, line 17.  "Non directly cytopathic" has been changed into "non-cytopathic".    
